# Natural Polymers and Their Nanocomposites Used for Environmental Applications

**DOI:** 10.3390/nano12101707

**Published:** 2022-05-17

**Authors:** Ecaterina Matei, Andra Mihaela Predescu, Maria Râpă, Anca Andreea Țurcanu, Ileana Mateș, Nicolae Constantin, Cristian Predescu

**Affiliations:** Faculty of Materials Sciences and Engineering, University POLITEHNICA of Bucharest, 060042 Bucharest, Romania; ecaterina.matei@upb.ro (E.M.); anca.turcanu@upb.ro (A.A.Ț.); ileana.mates@upb.ro (I.M.); nctin2014@yahoo.com (N.C.); cristian.predescu@upb.ro (C.P.)

**Keywords:** polymers, nanocomposites, heavy metals, PM, soil remediation

## Abstract

The aim of this review is to bring together the main natural polymer applications for environmental remediation, as a class of nexus materials with advanced properties that offer the opportunity of integration in single or simultaneous decontamination processes. By identifying the main natural polymers derived from agro-industrial sources or monomers converted by biotechnology into sustainable polymers, the paper offers the main performances identified in the literature for: (i) the treatment of water contaminated with heavy metals and emerging pollutants such as dyes and organics, (ii) the decontamination and remediation of soils, and (iii) the reduction in the number of suspended solids of a particulate matter (PM) type in the atmosphere. Because nanotechnology offers new horizons in materials science, nanocomposite tunable polymers are also studied and presented as promising materials in the context of developing sustainable and integrated products in society to ensure quality of life. As a class of future smart materials, the natural polymers and their nanocomposites are obtained from renewable resources, which are inexpensive materials with high surface area, porosity, and high adsorption properties due to their various functional groups. The information gathered in this review paper is based on the publications in the field from the last two decades. The future perspectives of these fascinating materials should take into account the scale-up, the toxicity of nanoparticles, and the competition with food production, as well as the environmental regulations.

## 1. Introduction

The use of polymers as sustainable products for life quality has been intensively studied and successfully applied for many years. The well-known polymer products for pharmaceutical and biomedical applications in controlled drug release or the novel methods for polymer-based pharmaceuticals have been successfully used [1]. Polymers have a central role in modern life. Moreover, the recent development in nanotechnology offers new ways to develop nano-sized materials with a high impact on biomedicine, food, and environment applications. In particular, polymeric nanocomposites bring an interdependence between matrix and reinforcement materials; thus, excellent advantages appear by this scientific approach. In this way, polymeric nanocomposite fabrication is a part of polymer nanotechnology. 

An important aspect consists in the compatibility between two phases (continuous—matrix and discontinuous—reinforcement). In the case of the dispersion of the nanoparticles, reinforcements in polymeric matrix agglomeration and possible clustering have to be avoided, [2]. Obviously, the choice of the nanoparticle types is linked with the applications and targeted properties of the polymeric nanocomposites. There are many known synthesis techniques of polymeric nanocomposites, such as the sol-gel process, co-precipitation, chemical reduction, reverse micellar synthesis, microemulsion, the hydrothermal process, laser pyrolysis, or laser ablation [2]. 

Today, the progress in polymer production is also based on natural polymer applications as green solutions for a clean environment. Polymers such as starch, cellulose, and poly(lactic acid) (PLA) are integrated into cosmetic products, and their biodegradability helps with the diminishing of disposal areas. Regarding the waste solubility and weak stability of naturally derived polymers, the chemical modification of these permits the achievement of remarkable properties. An example could be the results obtained using simulated compost environments where cellulose acetates revealed biodegradability and the degrees of substitution were up to 2.5 [3]. Green chemistry considers renewable sources as sustainable solutions for advanced and efficient materials. In this context, agro-industrial waste derived from natural substances assures stability for ecosystems and a low carbon footprint and is in accordance with the circular economy concept. The challenge still remains to obtain valuable products from renewable resources with biodegradability or compostability potential as an alternative in the case of their end of use [4].

It has been defined that natural polymers (associated with biopolymers) result from the metabolism of living organisms and represent monomeric units that form macromolecular structures through covalent bonds [5,6,7]. They perform vital functions in nature, being responsible for the preserving and transmitting of genetic information and the storing of cellular energy. Their main advantage is their biodegradability, whereby CO_2_, although it is released, is rapidly and directly absorbed by agricultural crops and soil. Most of these biopolymers are of the polysaccharide class, such as cellulose (found in about 33% of all plant components), chitin/chitosan, starch, and lignin [6,8].

As a short review of these mentioned polymers and their nanocomposites, this paper identified some natural sources for these types of advanced materials and their applications, as is presented in Figure 1. 

The usual polymers, such as polyethylene (PE) and polypropylene (PP), have a long period of stability, and their degradability takes place over a few years. Due to this disadvantage, the pollution of the environment can increase when using synthetic polymers. The use of these natural polymers can represent a valuable solution.

The novelty of this paper is to emphasize the main performances obtained when natural polymers are used as a matrix for the development of advanced nanocomposites for environmental decontamination. Thus, the most used natural polymers obtained from natural sources, as single materials or combined as composites, are presented for water–soil systems or air decontamination when pollutants such as heavy metals, organics or inorganics, and microorganisms should be removed. 

According to the literature research, the most promising and commonly reported natural polymers, as single materials or in combination with other advanced materials, are chitosan, alginate, cellulose, lignocellulose, starch, and PVA. The main properties of biopolymers to be used as filter membranes or adsorbents for environmental applications are related to their high permeability for capturing airborne or water contaminants [9], the outstanding mechanical and thermal properties [9], better resistance to diverse pH conditions [10], biodegradability, swelling capacity, the slight modification needed to increase the adsorption, and last but not last, the low cost [11]. For the processing of fibers by electrospinning technology, the specific features of biopolymers are needed for the adequate solubility and compatibility in aqueous solvents, a certain molecular weight, and a desired surface topography to attract positive or negative ions. All of the biopolymer requirements will permit the obtaining of nanometer-sized fibers with a controlled fiber diameter, without bead defects, and with a specific surface area [12]. In addition to medical applications, environmental factors could also be remediated by the use of these natural materials. It is well known that sustainable materials are usually derived from synthetics due to their durability, but this causes a great threat to the environment due to their harmful chemicals. In recent years, it has been shown that replacing at least one synthetic component with a natural one could enhance environmental protection without compromising the original properties of these materials. In this regard, our new approach sustains the use of these natural polymers as a viable alternative for a clean and safe environment. 

Based on our study, the published research from the last 20 years has focused on the retention of heavy metals as target pollutants in water and their immobilization in soil by natural polymers and their nanocomposites as adsorbents, foil, membrane, hydrogel membrane (HM), and metal–organic framework (MOF) shapes. With regard to air as an environmental factor, the literature offers solutions especially for the retention of particulate matter (PM). To a lesser extent, the studies have also focused on organics removal. All this information presented in this review paper has been classified according to the importance derived from the data analyzed. Even though this review is restricted to articles published in the last two decades, also presented are older but relevant data related to the natural polymers.

## 2. Natural Polymers as Sustainable Advanced Materials for Environmental Protection

The structural complexity of carbohydrates, plant materials, and bacterial biomass offers a large quantity of natural as well as monomeric feedstocks polymers [4]. Among natural polymers, polysaccharides are the most widely applied biopolymers, which are environmentally friendly and also used for medical applications [13]. Moreover, the highest quantities of natural polymers on earth are represented by lignin, followed by cellulose [14].

Polysaccharides generally originate from plants and consist of a minimum of two monosaccharides which are covalently bonded by glycosidic links. They are mostly used in the food and pharmaceutical industries and are water-insoluble at a natural pH, but because of their hydrophilic functional groups, they have a great capacity to absorb water. At the same time, the wettability, cross-linking density, and polymer structure flexibility represent important characteristics that influence the water absorption efficiency of the polymer nanocomposite [15]. 

Chitin and chitosan are easy to process in various forms; so, obtaining advanced nanometer-sized structures with a large specific surface area and porosity that would allow the retention of considerable amounts of heavy metals compared to micron particles has led to the obtaining of nanostructures with high adsorption capacities from both water and modified contaminated soil. Chitin is the second major component in nature, after cellulose, and it is obtained industrially from marine sources in quantities of about 1 billion tonnes. The sources could be different types of fungi, arthropods, mollusks, mushrooms, algae or fish, and some plants [16]. This polymer is the main source for most biodegradable plastics [8]. It exhibits essential properties in obtaining materials, with applications in medicine, the pharmaceutical and food industries, cosmetics, and the environment, as it is a powerful antioxidant, anti-inflammatory, and skin regenerator, as well as an adsorbent or ion exchanger, especially when the dimensions are nanometric, and its structure is crystalline [7,17,18,19]. Chitosan is generated from the waste of the biomass and fishery production [7]. 

Chitosan represents a good candidate for heavy metal removal from waters. Chitosan is produced by chitin deacetylation. Using chitosan in combination with nanoparticles can solve the problem of gelation and agglomeration [20]. For example, montmorillonite and bentonite represent some of the most used clay minerals in combination with chitosan for Ni, Pb, and Cr removal from waters and contaminated soils [21,22]. Their efficiency is strongly dependent on the pore structure, pH, and surface area. Chitosan (based on deacetylated chitin with a degree of deacetylation ≥70%) is a biodegradable biopolymer extracted from waste from various sources, such as shellfish and fungal biomass; it is non-toxic to humans, antimicrobial, environmentally friendly, and biocompatible and is intensely and successfully applied in medicine, cosmetics, and heavy metal biosorption [23,24,25,26,27]. Because of its intense use, green synthesis was particularly developed when nanoparticles were capped into the chitosan substrate [28]. Regarding the advantage of using chitosan for the retention and immobilization of heavy metals in polluted systems, this process is possible through amino and hydroxyl functional groups [29,30,31,32]. In solid form, chitosan and its derivatives can be used as soil amendments and for the immobilization of metals, especially in moist soil, considerably reducing the toxicity in living organisms from heavy metals [33,34,35]. Important applications of chitosan are also in the decontamination of wastewater from the mining or galvanizing processes [36]. The estimated price for chitosan was between 15 and 50 USD/kg [16]. 

Cellulose is the most widespread biopolymer; it can be present in nature or obtained from various natural materials (wood, cotton, and plant biomass). The tendency of cellulose particles to aggregate leads to the formation of microfibrils, which later can form cellulose fibers, which in turn can lead to a new class of composite materials, such as nanocellulose fibers with advanced properties [37,38].

Nanocellulose has a large specific surface area, high crystallinity and optical transparency, and stability in aggressive environments, while maintaining its biodegradability [39]. Nanocellulose fibers can be used as reinforcing polymers for nanocomposites, through hydrogen bonds forming stable structures with other nanomaterials. Moreover, nanocellulose could exist as nanocrystals with a crystalline structure and as nanofibrils with combined amorphous and crystalline regions [37]. In nature, cellulose nanofibers are found in the cell walls of plants, as a matrix of cellulose and hemicellulose [40]. Moreover, there are known so-called modified celluloses that are composed from regenerated cellulose and cellulose acetate (CA) [41]. Cellulose is the main component of plants, with an estimated production of about 7.5 billion tonnes/year [16]. Cellulose fibers are found in wood, bound by a binder, lignin. The extracellular enzyme complex cellulase, as a part of most microorganisms, acts on cellulose biodegradation [16,42]. CA is intensively used for electrospun nanofiber production. Often, in order to enhance the chemical and mechanical properties, when combined with reduced diameter fibers and increasing strength, some additives such as cationic surfactants are mixed with CA. In this way, the bead formation specific to CA nanofibers is diminished [43,44,45,46,47]. Most of these cationic surfactants, such as cetylpyridinium bromide (CPB), expose biocide properties and lead to nanofiber applications in the pharmaceutical industry [48,49,50,51] and in filtering airborne nanoparticles [43].

Hemicellulose is a polysaccharide formed with an amorphous structure and low mechanical and hydrolysis resistance, being incorporated into the cell walls of plants, and bound to pectin, forming a network of cross-linked fibers [16].

Another polysaccharide resulting from glucose/sucrose fermentation by Xanthomonas bacteria is xanthan, with good additive and rheological properties; it is used for cosmetics, as a stabilizer for advanced structures, etc. [16]. Lignin is a cross-linked polymer with a complex and heterogeneous structure; it does not belong to the class of polysaccharides, but it is found in nature, in chemical association with cellulose, giving lignocellulose, which is present in the cell walls of plants. It has industrial applications in biofuel generation and is obtained as a secondary product in the paper industry; it is a hygroscopic thermoplastic which possesses low deformability and a high rigidity [16]. 

Alginate is a biodegradable, biocompatible, renewable, non-toxic natural anionic biopolymer, obtained from various species of brown seaweed, and is a calcium, magnesium, and/or sodium salt of alginic acid. Its various applications are known, especially because of its gelatinous appearance, its ability to form hydrogels and to absorb water, and its metal ionic species; it is applied in medicine, environmental protection, cosmetics, and the food industry [52]. Alginate can be combined with ions such as calcium, which leads to the formation of three-dimensional hydrogels. Alginates are composed of α (1→4) linked L-type guluronic acid (block G) and β-(1→4) linked D-mannuronic acid (block M) block structures, which induce the ability to bind other ions (through block G) by developing resistant and rigid gels, respectively, with flexible and large diffusion rates (through block M) [16]. The Ca alginate form has an affinity for heavy metal retention by functional groups such as carboxylate, amine, phosphate, sulfate, and hydroxyl. The pH of the solutions influences the ability to bind to these functional groups from the alginate being usually in the form of balls or spheres, often functionalized with nanoparticles precisely to increase the ability to remove heavy metal ions from related systems [53]. 

PLA is a polymer derived from lactic acid and from renewable resources such as corn starch or sugar cane sucrose [16]. PLA is used in the form of films for packaging and the production of disposable containers for various biomedical applications due to its biocompatibility and biodegradability [3]. PLA first undergoes abiotic hydrolytic degradation, and then, microorganisms (mainly bacteria and fungi, which form biofilm) metabolize the lactic acid and its oligomers are dissolved in water.

Silk represents a unique natural material, the fibers being obtained from the silk cocoons of B. mori [54]. Its high mechanical strength and biocompatibility, especially cytocompatibility in vitro, imposed its use as a biomaterial for the medical device industry and for regenerative medicine [54]. Tunable properties facilitate its application for materials such as facial masks for air filtration.

In addition to these natural polymer materials, an important class for environmental remediation is extracellular microbial substances, identified as extracellular polymer substances (EPSs) [55,56,57]. This mixture of polymers resulted from micro-organism secretion after various kinds of biochemical consumption; they are classified as soluble or bound EPSs [55]. EPSs have potential as bioflocculants and adsorbents for various pollutants. 

## 3. Polymeric Nanocomposites as Sustainable Advanced Materials for Environmental Protection

Polymeric nanocomposites consist of polymers reinforced with small quantities of nanoparticles, which lead to the advanced properties of new materials. These materials represent sustainable alternatives in comparison with conventional polymers for modern society, where thermoplastic, thermoset, or elastomer materials are essential in high quality applications [58,59]. 

In general, composites contain three main types of matrixes, polymer, metal, and ceramic, combined with various additives as reinforcement (fillers, fibers, flakes, and/or particles). In this context, nanocomposites contain at least one nanoscale component [59,60]. This component may have different geometries, leading to linear nanocomposites (one-dimensional such as carbon nanotubes), two-dimensional laminate, (such as montmorillonite), or three-dimensional powders (such as silver nanoparticles). These geometric features induce the different functionalities of the formed nanocomposite as a whole [60].

However, the importance of some eco-friendly material production, without an unsafe impact on the environment, is compulsory today. One solution is the use of renewable sources instead of those which are synthetic. Thus, waste disposal and energy consumption are diminished instead of there being green natural material use and degradable processes for the final products.

Biodegradable polymers are modified by microbial populations that lead to mineralization. Parameters such as pH, humidity, oxygen, and metal content are constantly monitored in the biodegradation process [60,61]. There are various natural polymers, such as starch, cellulose, PLA, and lignin, which can be matrices for various nano-sized fillers, thus leading to nanocomposites obtained by techniques such as templates, interleaving in solution or melting, and in situ synthesis [62,63,64].

It is well recognized that the toxicity of nanoparticles, as well as the methods used for their synthesis and modification with natural polymers, can have a negative impact on the environment. The impact of the nanoparticles is larger than that of the polymer matrices. The addition of NPs to the natural polymer matrix permits the obtaining of nanocomposite polymeric materials which are beneficial for the environment in that they are related to replacing the limited petroleum resources and the valorization of sustainable resources for the obtaining of natural plastics, as well as the creation of the performing properties for different environmental applications [65]. For example, the chitosan/silver nanocomposite showed better enhanced antimicrobial activity than chitosan by itself in wastewater and the decolorization of methyl orange (MO)—Figure 2.

An extensive classification of nanocomposites was achieved by Zaferani in his paper, where the properties and challenges of these materials are explained [60]. Thus, two major classes are presented: (i) polymer-nonmetallic and (ii) polymer-metal nanocomposites, with the main characteristics. Among these, for the first class, polymer-carbon nanotubes, polymer-graphene, or polymer-clays are the most significant materials. For the environmental remediation, nano-clays expose eco-friendly features and low-cost production. Due to the cross-linking effect, the mechanical and permeability properties are enhanced. One important aspect consists in the good dispersion and proper intercalation of nano-clay layers in order to obtain envisaged performances [60,67]. Moreover, for polymer-metal nanocomposites classes, metal nanoparticles as a reinforcement phase display a high surface area; thus, reactivity is increased with the decrease in size. Silver nanoparticles are intensively applied in biomedical and environmental applications due to their antibacterial properties. Other important nanoparticles integrated into the polymer matrix are gold and palladium [60]. However, the toxicity level of polymeric nanocomposites induced by metal nanoparticles should be investigated as they show some potentially adverse effects on biota [68].

Chitosan-modified zeolite composites are a promising platform for environmental engineering applications [69]. The synergetic impact between the chitosan with zeolite nanoparticles led to a biocompatible mesoporous network material with low toxicity, improved mechanical properties, narrow pore-size distributions, and high surface area features as adsorbents of water pollutants.

Inspired by natural zeolites, the most interesting and important types of nanoparticles that have been dramatically exploited for environmental purposes (air and water purification) are metal–organic frameworks (MOFs) and their popular subclasses (zeolitic imidazolate frameworks (ZIFs), Materials Institute Lavoisier frameworks (MILs), etc.). MOFs are wonderful materials, widely studied for removing contaminants from the effluents [70]. They are characterized by nontoxicity, high chemical stability, high adsorption capacity, large porosity, high inner surface area, and ultrahigh thermal and chemical stabilities [69]. Nanocellulose-based filtering materials were developed by the incorporation of ZIF nanocrystals in the cellulose microfiber network, as a subclass of MOF HKUST-1 [9]—Figure 3.

The addition of ZIF-8 nanocrystals to cellulose microfibers led to an improved surface area, proved by an excellence enhancement in the BET surface area from 6.66 up to 620.80 m^2^/g, compared with the unmodified surface, which led to an increase in the filtration efficiency from 99.5 to 99.9% against PM 0.3 particles [9]. ZIF-8 crystal was found to coat the surface of the chitosan/polyvinyl alcohol electrospun nanofiber (CS/PVA-ENF) for dye removal from wastewater treatments [71]. In this case, the high adsorption capacity of 1000 mg/g was recorded during the second cycle.

In another paper, zeolitic imidazolate frameworks-67 (ZIF-67) crystals were incorporated on the surface of a magnetic eggshell membrane (Fe_3_O_4_@ESM), resulting in the ZIF-67@Fe_3_O_4_@ESM composite as a novel adsorbent with the high surface area of 1263.9 m^2^/g [70]. The maximum adsorption capacities of 344.82 and 250.81 mg/g for Cu(II) and Basic Red 18 (BR18) dye, respectively, were reported [70]. The advantage of the use of this adsorbent is that the magnetite favors the facile separation from aqueous media. Recently, MILs have attracted much research attention as they are promising for the adsorption of heavy metals removed from wastewater [72]. 

Natural polymers and their combinations are successfully used for the creation of hydrogel membranes (HMs), a cross-linked three-dimensional (3D) porous structure that can act as an adsorbent of heavy metals or organic contaminants from treatment wastewater. For example, HMs based on PVA for the adsorption of strontium ions from wastewater treatment; calcium alginate (CaAlg) coated with iron nanoparticles (Fe NPs, ~5 nm) having a 99.5% efficacy to remove Cr(VI) from contaminated water [73]; magnetic chitosan cross-linked with glyoxal (Fe_3_O_4_NPs/CS/glyoxal) for the removal of 80–90% of Cr(VI) from water [74]; and carboxymethyl cellulose (CMC) g-poly(2-(dimethylamino) ethyl methacrylate) (CMC-g-PDMAEMA) to remove MO from aqueous water with a high adsorption capacity of 1825 mg/g [75] are reported.

It has been proven that polymeric nanocomposites could represent an increasingly significant role for enhancing the environmental factors with respect to the removal of heavy metal ions and organics from waters and soil [76].

## 4. Remediation of Water/Soil Systems

Heavy metal pollution represents one of the most significant environmental issues, with a harmful effect on biodiversity, due to their persistence, nonbiodegradability, and biomagnification effects in the whole trophic chain. For example, metallurgical processes are still one of the major pollution sources, and the soil is the main factor that contributes to mitigation through groundwaters, vegetation, and large land areas. High quantities of Cu, Pb, As, Cr, and others are leached together with sulphuric acid; thus, the remediation technologies are continuously updated in order to assure proper quality of the environmental affected factors [77]. 

The possibility of polymer use as biodegradable macromolecules for heavy metal immobilization from soil has been intensively studied; generally, the use of polymers for soil is focused on their mechanical strength and reinforcement properties for soil durability [78]. Most of the studies were developed at a laboratory scale, and leaching tests were applied to the theoretical data being acquired. It is important to emphasize that polymers have a significant impact on pH values for leachate. As a rule, the concentration of heavy metals decreases with soil depths and with polymer addition, leading to groundwater protection.

Compared with organic pollutants, heavy metals are persistent, and their degradation is difficult to achieve [79]. Their persistence is highlighted in different forms: adsorbed on soil surface or chelated with organic matter, as oxides or hydroxides, as well as in organisms or residues, and soluble species in water (as ions or chelates structures) [80]. 

The advantage of polymer use for heavy metal capture is their reuse after concentration and also the regeneration of the polymers. However, the high efficiencies of polymer applications also involve high costs with regard to ultrafiltration costs and the membrane clogging risk [81]. In addition, calcium and magnesium ions are also affected. Today, tunable polymers are developed where natural polymers have become more efficient through other types of polymers or modified inorganic particles. Moreover, the flush process of soil could be applied, and the soluble heavy metals are transferred to a liquid phase and adsorbed onto polymers added to the liquid phase. The polymers are then separated by centrifugation or sedimentation and regenerated [81]. Polysaccharide nanocomposites represent the major class of nanocomposites derived from the natural polymers that have attracted considerable interest [82].

Most metals play a key role in the functioning of living organisms, in different quantities. However, the extra growth of both the essential (Zn, Cu) and the non-essential (Cd, Hg or Pb) species can cause chronic or acute conditions and can lead to the spread of effects throughout the food chain [83]. Moreover, non-essential metals, such as mercury, lead, and cadmium, which are constantly occurring from industrial activities, can be bioaccumulated, presenting the risk of adsorption and thus difficult removal from the affected areas. Most trace metals participate in adsorption reactions developed at the groundwater interface; so, their removal can be conducted by binding to other natural polymeric macromolecules (such as humic substances or bacterial polymers) or colloidal natural particles (such as clay, microorganisms, and biological matter); some are even dispersed in groundwater, becoming carriers for target metals and helping to concentrate and subsequently separate them from systems [83,84].

In Table 1 are presented the performances obtained using chitosan as a matrix for advanced nanocomposites used in the decontamination processes for the water–soil systems. An interesting issue is its double functionality demonstrated by the simultaneous testing of both water and soil. 

Regarding the heavy metal ion contamination of water, the researchers are making continuous efforts to find innovative and cost-efficient solutions in order to solve these problems. The removal of these ions is possible using different techniques that can sort the target species based on their size, their volatile or soluble properties, or their chemical reactivity [85,86]. The polymers represent one of the emerging classes of materials for water treatment, i.e., for the retaining of metal ions. They share specific functional features that can be adapted to meet the demands of a broad range of wastewaters. The polymeric nanocomposites represent advanced materials that provide an improved performance thanks to the integration of both polymer and nanomaterial properties. Due to the fact that the conventional treatment technologies proved to be not so efficient and or expensive, the focus was attracted by the use of polymers for water remediation. 

In addition to heavy metals, water could contain considerable organic pollutant quantities with a high impact on life quality. Some of these emerging pollutants present serious threats due to their toxicity and lack of regulation. The organic pollutants with the greatest impact on the environmental factors are dyes, pharmaceutically active compounds, endocrine-disrupting chemicals, personal care products, and flame retardants [87]. 

In accordance with the required legislation limits, the water management focuses on the improvement of the aquatic ecosystem quality and environmental protection even though official limits for these type of pollutants are still not available [88].

Together with these, polycyclic aromatic hydrocarbons (PAHs), persistent organic pollutants (POPs), especially dioxins, furans, and chlorinated pesticides (OCP), and polychlorinated biphenyls (PCBs) are also monitored both in water and soil as they are present on a large scale in ecosystems due to human activities [89].

There are various treatment methods for the removal of organics, among them the most used is adsorption with the help of nanomaterials, nanocomposites, nanoparticles, clays, biopolymers, metal–organic frameworks (MOFs), and zeolites [90]. 

They can be used as natural or nanocomposite polymers, and they have the advantage of being effective and inexpensive. Furthermore, besides their extensive application as flocculants or coagulants, they can be also applied in membrane systems for water decontamination [91]. CA, polycarbonates, polyethylene, chitosan, and alginate are the most applied polymers for membrane technology. A review of recent research regarding the use of polymer membranes in water treatment was performed by Khodakarami and Bagheri (2021) [92]. These kinds of materials have been widely applied in order to avoid the membrane blocking in several filtration methods. The grafting of polymer chains on membrane surface represents one of the most used methods for improving the membrane performance [93]. Extracellular polymers contain polysaccharides and proteins in exopolymers. Due to their acidic nature, these anionic polymers can easily bind metals. At acidic pH, if extracellular polymers are present in the soil, they can increase the adsorption of metals on the surface. In the case of alkaline pH, dissolved bacterial polymers can bind the metals in traces in the aqueous phase, which reduces the metals in the soil. Some examples are those for Cd^2+^, with the formation of stable complexes with N and S, and those for Pb(II), with O, N, and S [94]. Table 2 comprises the most significant performances of natural polymers and their nanocomposites presented in the literature regarding water decontamination.

Regarding heavy metal removal (Pb, Hg, Cd, Ni, Zn Fe, and Al), especially from industrial effluent, the bacterial polymers originated from *Pseudomonas*, *Halomonas*, *Paenibacillus*, *Bacillus*, and *Herbaspirillum* have the ability to be used as flocculating agents for their removal. Moreover, these polymers could be extracted from activated sludge, wastewater, and other sources [56]. In the case of wastewater with a high N content, the high quantities of microbial polymers could be formed, and high efficiencies for the immobilization of heavy metals were registered. Siddharth et al. [55] presented in their review paper the types of these microbial polymers produced by different bacterial species that act as bio-adsorbents for heavy metal removal (*Bacillus licheniformis* for Cr (VI); *Bacillus mucilaginosus* for Fe and Pb; *Herbaspirillium* sp for As, Zn Mn, Al, Fe, Pb, and Cr; *Cloacibacterium normanense* NK6 for Al, Cu, Ni, Fe, and Zn, and *Rhizobium radiobacter* and *Bacillus sphaericus* for Ni and Cu, etc.).

Another water parameter, such as turbidity or COD, could be investigated and removed by the use of bacterial polymers. For example, turbidity removal from raw water has been reported by using bacterial polymers from *Bacillus Spas* at 86% [131] and EPSs synthesized from *Bacillus licheniformis*, with CaCl_2_ added to drinking water, leading to a 95.6% removal of turbidity and a 61.2% removal of COD.

In the case of wastewater with a high content of suspended solids, a possible ionic linkage between bacterial polymers and multivalent cations (Mg^2+^ and Ca^2+^) could support a bacterial flocculation process, together with flocs formation due to the compression of the ionic layer and the aggregates formation [132,133].

The research studies indicate that the bacterial polymer flocculation capacity increased with an increase in protein content, and the low concentration of humic substances increased with the number of total bacterial polymers because each component of these polymers contributes to the overall efficiency of the flocculation process.

The application of these bacterial polymers as coagulants in order to replace classical inorganic coagulants (alum or ferric salts) represents a promising alternative for water and wastewater treatment processes.

Soil can be affected by pollution with heavy metals or radionuclides, having a severe consequence on ecosystems and human health [84,134,135,136,137]. ^137^Cs, ^90^Sr, and uranium are the most known radioactive isotopes, which present a significant danger for the modern world [138]. Moreover, the presence of nanomaterials and bioplastics in soils requires specific materials to be able to entirely biodegrade the polymers in the soil and thus maintain the ecological restoration and soil quality.

Among the most polluting are the heavy metals, which can be adsorbed with high efficiency when modified biomass such as biochar is applied. The adsorption process takes place through surface complexation, followed by precipitation and ion exchange and/or physical adsorption [139,140,141]. There are various methods for heavy metal soil remediation, such as stabilization, excavation, bioremediation, landfill, and soil rinsing techniques [142,143,144,145], where various materials are used as organic amendments, such as biosolid compost and biochar [146,147]. Moreover, biochar is a promising resource for the acceleration of the degradation of polyhydroxybutyrate-co-valerate (PHBV) and its composites containing AgNP in soil [148].

Among these, natural polymers are often used for their agro-environmental compatibility and efficiency [149,150]. Table 3 shows some significant results reported for soil remediation using natural polymers and their nanocomposites.

One of the most analyzed materials is biochar as a potential adsorbent for heavy metals and the restoration of soil quality. The stability, the performance, and the facile use are sustained by its combination with chitosan as a natural polymer and MgCl_2_ when an advance composite based on magnesium oxide biochar–chitosan is applied and used for Cd removal. Here, chitosan acts as a chemical adsorbent and an ion exchange resin for Cd (II) removal from soil and also from water (Figure 4) [84].

Chitosan as a natural polymer is often used in water and soil decontamination, especially when the graft-polymerization process is applied in order to enhance its properties. For example, montmorillonite-rich bentonite grafted with chitosan as an inexpensive and sustainable composite has an immobilization capacity for heavy metals (Cu, Zn, Cd, and Ni) from soil [22]. In this case, two natural sources are used: chitosan as a biopolymer obtained from seafood wastes and bentonite as an abundant mineral. The presence of biopolymer in the composite structure increased the adsorption capacity of the composite used for soil and water. Another advantage of the chitosan as a biopolymer consists in its immobilizing capacity when used for soil remediation. Due to the availability of the adsorption sites, Cu exposes a higher affinity for the composite surface in comparison with Zn, Cd, or Ni. The binding capacity of the adsorbent for heavy metals was also studied through desorption studies where the results indicated higher values for all metals [22,157].

The porous structure of the composites based on natural polymers enhanced the adsorption capacity, and graphene use increased the mechanical strength of this. Thus, a carboxylated graphene oxide/chitosan/cellulose (GCCSC) bead composite was used for Cu(II) removal from both water and soil. Chitosan was used for bead formation, combined with the cellulose that offers strength to the beads [85].

Montmorillonite-supported CMC-stabilized nanoscale iron sulfide (CMC@MMT-FeS) was used for soil remediation in order to immobilize Cr(VI). Thus, an Fe(III)–Cr(III) complex was formed after 30 days. The authors indicated that the acid-exchangeable fractions of Cr from the soil were converted to oxidable and residual fractions [155].

CMC was used in soil remediation by integration into some nanocomposites: CMC-stabilized nanoscale zero-valent iron CMC-nZVI and CMC-stabilized nanoscale zero-valent iron composited with biochar CMC-nZVI/BC for the in situ remediation of Cr(VI).

The polymer and nanoscale iron integrated as nanocomposites led to a diminishing of the leachability for Cr_total_ or Cr(VI) in the affected soil by over 95% [156].

CMC, as a polysaccharide derived from cellulose, plays an important role as a dispersant for nanoparticle solutions, and its use as a matrix for FeS nanoparticles, combined with bone-char particles, provides a stable composite that could be loaded with phosphate-solubilizing bacteria. This advanced composite helps in lead passivation and the immobilization process in soil. The passivation process takes place by chemical precipitation, complexation, electrostatic attraction, and biomineralization, when stable structures such as Pb_5_(PO_4_)_3_OH, Pb_3_(PO_4_)_2_, and PbS were formed [161].

A composite based on calcium alginate matrix with magnetic properties for easy removal and AC was designed for 12 PAHs removal. Even though the removal was significant, low recovery results were registered, demonstrating the difficulty of the material’s use [157].

Alginate spheres with a magnetic hollow carbon composite (MHCC) were applied for free Cd^2+^ ion adsorption from the liquid phase of soil into porous alginate spheres and Cd desorption from the solid phase of soil. This composite represents an alternative for the in situ remediation of soil heavy metals, with low costs, efficiency, and natural resource availability [158].

In addition to the environmental issues, the release of nutrients such N in soil has to be integrated within the plant growth needs [152,162,163]. In this context, different formulations of urea increase the soil microbial activity compared to the conventional one, such as polyolefin-coated controlled release (CR) urea for maize crops or encapsulated urea into chitosan polymer for potato growth, due to the N content increasing [164]. It was observed that the great amounts of nitrogen-cycling microbial communities for potato crops are affected by the proper controlled release of N nutrient using a chitosan polymer–urea encapsulated fertilizer [152].

Chitosan, as one of the most abundant natural biopolymers, is well known for its environmental applications, especially for heavy metal removal from water. Due to its amino and acetamido functional groups, chitosan acts as a cationic polyelectrolyte character as it is involved in the chelation process for heavy metals. Together with nanoclays (for example modified and unmodified montmorillonite) and biochar as additives, the new composite material exposes strength, stability, and good adsorption capacity through the biochar porous structures and immobilization capacity by the NH_2_ active groups of chitosan for Cu, Zn, and Pb from acidic mining soil. By a leaching test, good efficiencies of heavy metal immobilization were recorded [154].

Fungal chitosan as nanoparticles could be produced from different biomass sources, such as *Cunninghamella elegans*. Compared with chitosan, this nanomaterial, as a nano-fungal chitosan, presented good adsorption efficiencies for heavy metals, such as Pb^2+^ and Cu^2+^, from water and contaminated soil due to its hydroxyl and amino groups and phosphoric groups resulting from a cross-linkage with sodium tripolyphosphate [23]. The immobilization process led to metal ion leaching and bioavailability, for both contaminated water and soil, by complex stabilization and adsorption, ion exchange, and/or surface precipitation [165].

One of the major applications of natural polymers is chemical soil stabilization in order to raise mechanical properties, together with permeability and stability [166,167]. Usually, these polymers are water-soluble, for example as polysaccharides (as natural polymers) or polyacrylamides (as synthetic polymers). Even though Portland cement has good stabilization properties, these polymers are eco-friendly materials with regard to carbon dioxide emissions, natural resources, and energy consumption [168]. Additionally, the natural polymer waste resulting from the pulp and paper industry, and fly ash as lignin, and from the food industry as polysaccharides could be reused for soil stabilization [169].

Due to their particle sizes, these polymers expose high specific surface area and variable surface charges and could be applied to soil to enhance physical and chemical soil properties. Thus, the mechanisms responsible for clay minerals and polymers are based on electrostatic interactions between cationic polymers and the negative charges of clay minerals. Anionic polymers act as flocculants, and the electrostatic interactions between these polymers and clay minerals depend on pH and are completely in the presence of polyvalent cations. In the case of uncharged polymers, an adsorption phenomenon appears on the clay mineral surface (as colloid), and high molecular weight polymers are adsorbed on the clay surface with a low desorption rate [14]. Another type of interaction could appear between polymers and high-dimensioned soil particles (sand, for example), where a thin polymer film is formed on the particles, and it is considered that a reinforcement mechanism of polymers takes place. A disadvantage of applied natural polymers onto soil represents their biodegradability, which could influence the endurance of the polymer in stabilized soils [166].

In order to improve soil erosion, polymers have gained attention since the early 1950s [170]. Thus, PVA as a natural polymer or polyacrylamide (PAM) were intensively used, and low quantities adsorbed on montmorillonite, quartz sand, and farm soil have led to improvements regarding structural stability [171,172]. For example, different soil aggregates sizes (lower than 1 mm and higher than 6.4 mm) were brought into contact with PVA as an anionic polymer. The optimum dose was 6.25 kg/ha, and the results indicate high efficiency in soil stability and water losses [170].

Polymers are components from organic natural compounds from aquatic environment, representing about 13% of the total COD in groundwater [83,173]. Moreover, the bacterial extracellular polymer production, intermediated by bacteria, influences the metal mobility in soil, exposing a high capacity for metal releasing, especially for Cd and Pb, at about a 2–4-fold increase in the presence of a polymer compared with Cd and Zn [174].

The bioavailability of Pb and Cd in the contaminated soils could be reduced by stabilizers based on natural polymers (lignin, CMC and SA), and a toxicity characteristic leaching procedure (TCLP) and sequential extractions were applied as methods for environmental impact assessment [159]. The oxygen-containing groups of the polymers act as chelators and immobilizers for Pb and Cd, with the leaching concentrations decreasing about 5.46–71.1% and 4.26–49.6%, respectively, in the treated soils. The contents of the organic forms of the two metals both increased with the increase in stabilizer dose on the basis of the redistribution of the metal forms by sequential extractions [159].

Microbial polymers are a high-molecular-weight mixture of polymers, initiating the binding through cohesion and adhesion of some polymers, such as polysaccharides and proteins, with cells [175]. These microbial polymers are soluble (as macromolecules or colloids in the growth liquid media) or/and bound (capsular, sheaths, and condensed gels) [55].

Soil quality and heavy metal immobilization could be achieved with the help of bacterial polymers, and the bioremediation efficiency depends on the biofilm resulting from the agglomeration of the polymer matrix and the bacterial communities [176,177]. The characteristics, such as nutrient accumulation, the protective layer onto the soil surface, sediment resistance, and the water-absorption properties lead to high efficiencies of soil bioremediation [56].

Sodium alginate has an efficient application when it is used as a substrate for the coating of FeSSi in order to form a nanocomposite with zerovalent nanoiron. The formed gel beads expose a high specific surface area, acting as biosorbents for Cd, Pb, Ni, and Cr [159].

The CA beads exhibited a high removal efficiency for the selective adsorption of Cu(II) when it was due to the exchange capacity with Ca^2+^; the heavy metal quantity was loaded onto alginate beads in the “egg box” structure [118].

The PVA/bentonite nanoclay/sodium alginate/iminodisuccinic acid (IDS) or 2-phosphonobutane-1,2,4-tricarboxylic acid (PBTC) nanocomposites were prepared by Toader et al. [138] by a casting method for surface decontamination of environmentally friendly water solutions. Their testing showed a decontamination efficiency (DF) in the range of 95–98% and 91–97% for heavy metals tested on a glass surface and the radionuclides ^241^Am, ^90^Sr-Y, and ^137^Cs on metal, painted metal, plastic, and glass surfaces, respectively.

## 5. Air Decontamination

Air pollution represents the main environmental threat for developing countries. Air filtration is still the most efficient and easily applicable air depollution technique [178].

Even though, during breathing, most of the particulate matter (PM) is blocked by the respiratory system, most PM 2.5, due to its size, goes through the respiratory tract [179,180]. The health risk is augmented by its high surface area, induced by the size leading to the possible adsorption of other hazardous compounds [43,180,181].

One of the most known atmospheric pollutants is black carbon (BC), with diameter sizes between 50 and 80 nm, produced during the combustion processes, especially when fossil fuels (coal, diesel, gasoline) and biomass are burning. The stability in the atmosphere affects the overall carbon balance, with a high impact on climate change and health quality [182,183,184]. This added to the recent pandemic problems when the virus, with sizes between 50 to 200 nm, was released as droplets by sneezing, coughing, or conversation, contributing to the overall atmospheric pollution [43,185]. Thus, more efficient filters have to be developed, including air conditioning systems. Moreover, microorganisms, together with other toxic gases, heavy metal dusts, and organic pollutants (polycyclic aromatic hydrocarbons, benzene, and aerosol) are mixtures included under the PM 2.5 classification with a high environmental impact [43].

Usual filters reveal low porosity (<30%) and are manufactured from porous materials as a substrate decorated with small pore sizes. Using nanofibers, a high porosity and surface area of the material appears, and a high efficiency of atmospheric pollutants is registered [186,187,188].

As main pollution sources, the automotive and aerospace industries still explore efficient solutions for the diminishing of air pollution. For this, new materials are the subjects of development and research as sustainable filters based on fibers, nanotubes, or different foams [189,190,191,192]. Among these, nanofibers are one of the most efficient and easily produced materials by electrospinning. Natural or synthetic polymers are a subject of the electrospinning process. The literature indicated the promising natural polymers to be polysaccharides, collagen, silk, and cellulose and the synthetic ones to be polyacrylonitrile (PAN), PLA, acrylonitrile butadiene styrene (ABS), polyurethane (PU), PVA, PEG, polystyrene (PS), PP, polyethylene terephthalate (PET), polyamide, etc. [193,194,195]. All these types of polymers could be electrospun and subsequently integrated into air filtration systems.

In order to achieve high performances for the filtration and separation process of pollutants, the selected materials have to possess the advanced characteristics suitable for pollutant removal. The mechanism of removal developed onto the material surface is in correlation with PM sizes, as is indicated in Figure 5 [196].

The main processes that can take place are impaction, interception, diffusion and electrostatic attraction.

Another important issue is the reuse capacity of the filters and their sustainability in relation to the environment. The development of new materials is still a challenge, but nanofibers and nanocomposites are promising structures with a high capacity for filtration.

Usually, the classical air filtration membranes, based on glass or other melted materials, consist of micrometer scale fibers [178,197]. Ultra-fine particles (PM 2.5) and bacteria are passed through these types of membranes due to their large pores, and the air quality remains affected [198,199]. When electrospun nanofiber membranes are used, the interrelated pore structure appears, which leads to a manageably sized pore with a high surface area and porosity [200,201,202,203,204].

One of the most important issues regarding air filtration membranes refers to their synthesis methods. Often, toxic organic solvents are used in preparation steps which could affect the environment. In addition, the solvents could be flammable, increasing the potential safety hazard. Another challenge is represented by the multifunctionality of the membrane; thus, inorganic, organic, and bacterial materials have to be simultaneously filtered. The basic synthetic polymers, such as polyimide (PI) [205,206], PU [198], PAN [207], polyamide [208,209], and polysulfone [178,210], have been successfully fabricated as nanofibrous membranes for filtration [211].

Not only air filtration efficiency sustains the membrane fabrication, but also its environmental impact; so, eco-friendly materials resulting from green synthesis are suitable for the overall membrane efficiency. In this way, green electrospun materials were developed to obtain fibrous membranes [212]. For this purpose, some natural and biocompatible polymers, or those that can be dissolved in nontoxic solvent, such as water, ethanol, or acetic acid, were developed and tested [213].

Thus, green methods such as electrospinning techniques with bio-based chitosan/poly (vinyl alcohol) nanofibers, including superhydrophobic silica nanoparticles for filtration efficiency, were developed. In addition to silica, the Ag nanoparticles were integrated into electrospun antibacterial nanofibrous membranes. These types of eco-friendly membranes revealed high performances and showed biological compatibility and antibacterial properties for PM 2.5; it has great potential application in eco-friendly air filtration materials, especially in personal air filtration materials. An overall look at the green electrospinning process, combined with UV treatment, is presented in Figure 6 [178].

One of the major disadvantages for this green electrospinning technology is the weak stability of the fibers, especially the nanofibers. This inconvenience is dealt with by thermal cross-linking and UV reduction technology. The Ag nanoparticle integration into the nanofibrous membrane exhibits antibacterial properties and high efficiency for non-oil and oil aerosol particle removal.

In addition, the pandemic disease enforces solutions for pathogen reduction, especially within the facemask production. It is well known that the facemasks are made from non-biodegradable synthetic materials, and with the huge quantity of their use, their disposal affects the environment. SiO_2_-Ag composite integration in a polymeric matrix (ethyl vinyl acetate) was developed as an innovative fabricated material with antibacterial activity towards *Escherichia coli* and *Staphylococcus aureus*, as well as towards SARS-CoV-2 [214]. Together with this problem, new solutions for sustainable and biodegradable materials as substrates for facemasks were developed. An example could be the renewable nanofibers [215]. Thus, hybrid composite nanofibrous layers were fabricated by the immobilization of TiO_2_ nanotubes as fillers into chitosan/PVA polymeric electrospun nanofibers. Chitosan/PVA and silk/PVA were used in this facemask filter as the middle and inner composite layers, with the roles of controlling protection and preventing contamination [216].

PMs combined with volatile organic compounds produce serious health problems. So, this inconvenience was resolved by the production of some efficient and eco-friendly air filters with high optical and multifunctional features. Basically, nanofibers forming silk protein, obtained by electrospinning, exceeded the conventional semi-HEPA filter efficiency, due to their optical properties. At the end of their use, these nanofibers are naturally degraded [217].

Moreover, poly(l-lactic acid) (PLLA) polymer as a biodegradable polymer derived from biotechnologies was used in order to obtain electrospun nanofibers for air filters. PLLA nanofibers indicated an efficiency of over 99% for PM 2.5, compared with a 3 M commercial filter. Thus, PLLA biodegradable nanofibers proved to have filtration capacities with a low cost and new perspectives for air industrial development equipment [218].

The main performances of polymer substrates for air depollution are presented in Table 4.

CA nanofibers combined with cationic surfactant CPB lead to nanofiber membranes by electrospinning is an efficient solution for aerosol nanoparticle and PM 2.5 removal from the air. The aerosols could be BC or coronavirus and 100% efficiency is achieved. The results indicate the possibility of future design for indoor air filters and facial masks using renewable and biodegradable polymers [43].

Multifunctional membranes based on chitosan as a natural polymer substrate combined with PVA and decorated with SiO_2_ and Ag nanoparticles were developed by an electrospinning process in order to obtain efficient membranes for air filtration [198]. This type of membrane also revealed biological and antibacterial properties [221]. Moreover, hydrophobic silica nanoparticles integrated into PVA–citric acid electrospun nanofibrous membranes indicated a high filtration efficiency [222]. A natural polymer KGM combined with an electrospun PVA nanofiber was tested for toxic particles from the air [221].

## 6. Conclusions and Future Perspectives

In conclusion, we highlight the recent performances regarding the use of natural polymers and polymeric nanocomposites, especially in the elimination and/or the immobilization of HMs and subsidiary organics from soil and water.

Enormous environmental threats, such as climate changes due to the carbon release, waste disposal, and water and air quality, force society to find sustainable solutions for life quality. Additionally, the well-known concepts, such as sustainable development combined with a circular bio-economy, have to be implemented such that biodiversity is not affected and future generations will have a stable and clean environment.

Available natural resources could be used as the next generation of the advanced materials with targeted applications. Moreover, agro-industrial biomass based on natural substances such as polymers, or mixtures of them, or biotechnology applied for monomer production offers interesting natural structures that could be tunable for enhanced properties. The application of natural polymers for the biomedical, pharmaceutical, and food industries is well known. In recent years, the research into environmental remediation, especially for soil, indicates promising results with natural polymers as single or nanocomposites, especially with chitosan. This paper integrates the most relevant results for water, soil, and air systems when natural polymers and their nanocomposites are applied as remediation materials. Based on our investigations, we observed that the combination of a natural component with a nanosized one led to the development of innovative materials with a real potential for the capture of the target pollutant from combined water–soil systems. The actual result proves the efficiency for heavy metal removal and opens new perspectives for the removal of organics based on the preliminary results. Our study demonstrates the advantages of using nanocomposites through the dual functionality of the two components (nanoparticles and polymers), which offer advanced properties, such as specific surface area, reactivity, and stability. In addition, the natural polymer as a green compound has the advantage of availability and low cost.

A more comprehensive vision for the future should be centered on scaled-up commercial and industrial applications. This will result from proven laboratory efficiencies combined with more and more environmental regulations.

## Figures and Tables

**Figure 1 nanomaterials-12-01707-f001:**
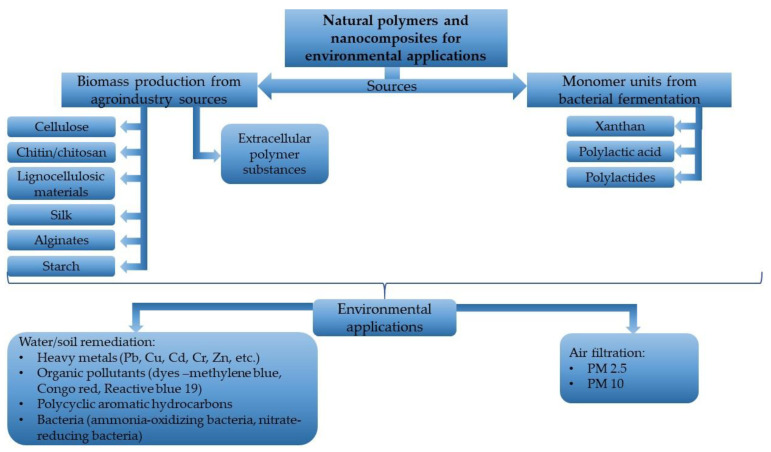
Natural polymers and nanocomposites for environmental applications.

**Figure 2 nanomaterials-12-01707-f002:**
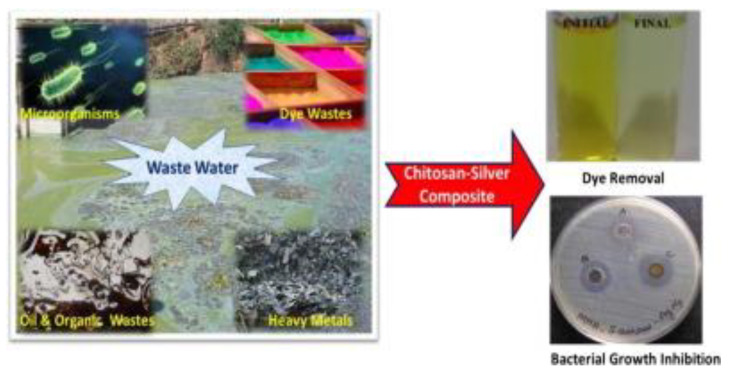
Eco-friendly approach of chitosan/silver nanocomposite used for dye removal for potable water. “Reprinted with permission from [66]. Copyright 2022, Elsevier”.

**Figure 3 nanomaterials-12-01707-f003:**
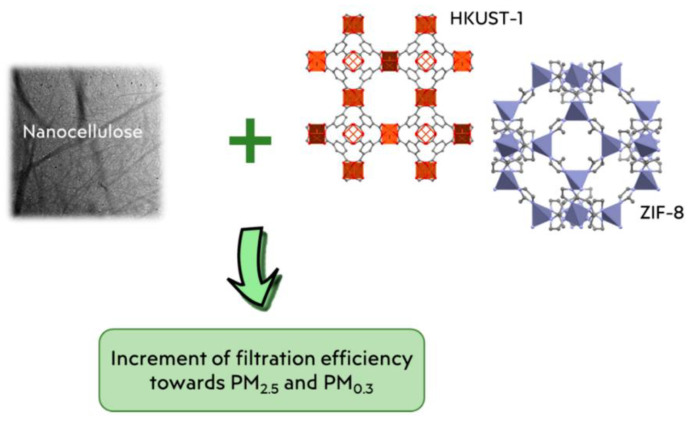
The performance of MOFs for air filtering media [9].

**Figure 4 nanomaterials-12-01707-f004:**
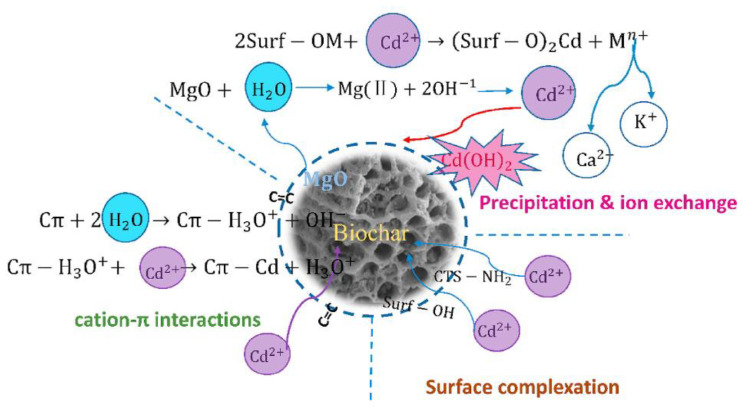
The mechanism diagram of Cd adsorption onto MgO-BCR-W [84].

**Figure 5 nanomaterials-12-01707-f005:**
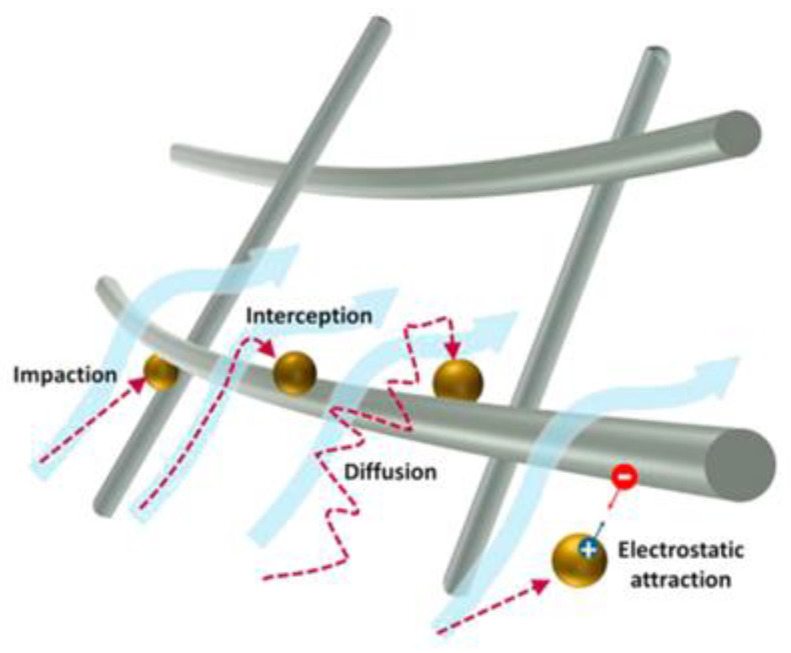
Four major types of particle filtration mechanisms: impaction, interception, diffusion, and electrostatic attraction [196].

**Figure 6 nanomaterials-12-01707-f006:**
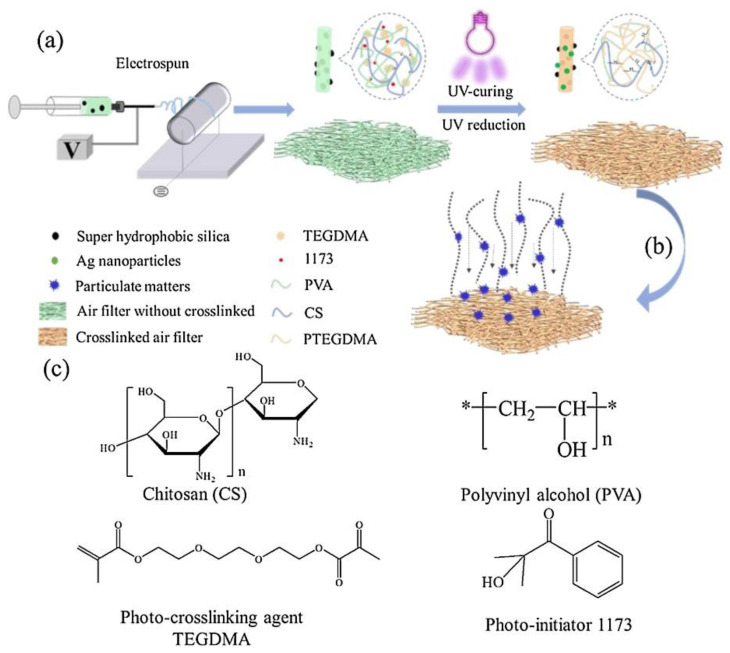
Example of green electrospinning process combined with UV treatment [178]. Fabrication process for antibacterial and hierarchical CS-PVA nanofibrous membranes by combination of (**a**) electrospinning, one step UV reduction and cured. (**b**) Filtration process of the CS-PVA@SiO_2_ NPs-Ag NPs air filtration membranes. (**c**) The chemical structure of CS/ PVA/TEGDMA/1173.

**Table 1 nanomaterials-12-01707-t001:** Chitosan and its nanocomposites with double functionality for water–soil system decontamination.

Type of Polymer or/and Nanocomposite	Water Performances/Mechanism	Soil Performances/Mechanism	References
Chitosan composite with magnesium oxide biochar (from rice husk), 2–22 nm.	59.66 mg/g Cd adsorption favourable, separation factor (RL): 0 and 2	2% composite: reduced Cd content bioavailable by 22.32%, Cd extractable in acid by 24.77%, and increased Cd residual by 22.24%.	[1]
Chitosan graft-copolymerized with montmorillonite rich bentonite	0.1 g composite with 25 mg/L metal solutions (50 mL), pH values between 1 and 8. % removal: from 60 to 95 for Cu(II), 35 to 95 for Zn(II), 20 to 85 for Cd(II)and 30 to 70% for Ni(II). Monolayer adsorption (Langmuir isotherm model), 88.5 mg/g (Cu), 72.9 mg/g (Zn), 51.5 mg/g (Cd) and 48.5 mg/g (Ni). Desorption with 0.05 M–ineffective, EDTA and DTPA - > 90% adsorbed metals removals.	1 g soil/0.1 g composite with metal retention capacity by 3.4, 3.2, 4.9 and 5.6-fold for Cu, Zn, Cd and Ni. The organic desorbing agents (ethylenediaminetetraacetic acid (EDTA) and diethylenetriaminepentaacetic acid (DTPA) > 90% adsorbed metals removal. [Ca(NO_3_)_2_] low desorption, suitable for metal immobilization. The Freundlich model described the adsorption process (N>1), metal adsorption capacity: 0.85 mg/g (Zn), 0.94 mg/g (Cu), 0.45 mg/g (Cd) and 0.42 mg/g (Ni). % Desorption lower for strongly adsorbed metal Cu (0.02% at 5 ppm to 0.27% at 50 ppm) than Zn (0.07% at 5 ppm to 3.03% at 50 ppm), Cd (0.2% at 5 ppm to 6.41% at 50 ppm), Ni (0.62% at 5 ppm to 5.58% at 50 ppm)—strong binding of metals by the chelating functional groups of the composite.	[2]
Nano-fungal chitosan nanopaticles (NCt) (from cross-linking with sodium tripolyphosphate) and chitosan Cts (from Cunninghamella elegans fungus), 5–45 nm.	Pb: 87.51 mg/g (300 ppm) and Cu: 89.12 mg/g (300 ppm).	0.25 and 0.5% NCt. Pb removal efficiencies for different samples: between 71.3 and 98.6%. Corresponding with bulk Cts: between 45.6 and 74.3%. Cu removal efficiencies for different samples: between 88.8% and 97.3%.	[3]
Composite carboxylated graphene oxide/chitosan/cellulose beads, about d = 2 mm.	22.4 mg/g Cu(II) for 40 μg/mL.	99.6% Cu(II) immobilization efficiency for 60 mg/L (88.6% for soil alone).	[4]

**Table 2 nanomaterials-12-01707-t002:** Natural polymers and their nanocomposites used for water pollutant removal.

Type of Polymer or/and Its Nanocomposite	Water Pollutants and Performances	References
Chitosan/clay nanocomposite by dip-coating technique, with the lowest pore size for ultrafiltration membrane: 13 nm	100% removal of 500 µg/L Hg(II) and 1000 µg/L As(III)	[95]
Chitosan hollow fibers/nanosized Fe_3_O_4_ as Fenton-like catalysts	89.4% removal of Reactive Blue 19 (RB 19) dye in continuous system and 74.4% for reused catalyst	[92]
Graphite oxide/poly(acrylic acid) grafted chitosan nanocomposite	Removal of dorzolamide (from pharmaceutical industry), 447 mg/g	[92,96]
Chitosan/gum arabic/carbon nanotube (CNT) as beads and membrane functionalized, BET surface area: between 78 and 198 cm^2^/g	Removal of solids from waters	[97]
Chitosan–montmorillonite membrane, with montmorillonite amount from 10% to 50% by mass.	Adsorbent for 80 mg/L Bezactiv orange V-3R dye, Qmax: 279.3 mg/g	[98]
Magnetic mesoporous carbon/β-cyclodextrinechitosan	Removal of fluoroquinolones, efficiency 90.7–99.7%, 130–165 mg/g	[99]
Glutaraldehyde cross-linked chitosan-coated Fe_3_O_4_ nanocomposites	Methylene blue (MB) removal, efficiency 96–98%, 758 mg/g	[100]
Chitosan/polyvinyl alcohol (PVA)/zeolite nanocomposite	Congo red removal, efficiency 94%, 5.33 mg/g	[92,101]
Quaternized chitosane organic rectorite intercalated composites	*E. Coli* removal, efficiency up to 90%	[102]
Chitosane–zinc oxide nanocomposite	Removal of permethrin, efficiency 99%	[103]
Polyethylene glycol (PEG)/chitosan nanocomposite	Removal of nitrates from waters, 50.58 mg/g	[92,104]
Chitosan/Al_2_O_3_/Fe_3_O_4_ nanofiber	Phosphates removal, 135.1 mg/g	[105]
Nano-SiO_2_-Cross-linked Chitosan-Nano-TiO_2_ nanocomposite	Removal of Hg, efficiency 98–99.5%, 1515.2 mg/g	[92,106]
EPSs as bioflocculant and bio-adsorbent (bacterial cells and natural polysaccharides, lignins, proteins).	1–10 mg/L EPS: 50% removal Pb(II) and Hg(II). 35 mg/L EPS: 72% Al, 40% Cu, 72% Fe, 85% Ni, and 45% Zn. *Bacillus licheniformis* as EPS strain used: 88% Cr (VI). *Cloacibacterium normanense* NK6: 73% Al(III), 36% Cu(II), 71% Fe(III), 85% Ni(II), and 65% Zn(II). *Herbaspirillium* sp: 26.6% As(III), 39.5% Zn(II), 31.4% Mn(II), 22.1% Al(III), 65.3% Fe(II), 25% Pb(II), and 94.9% Cr(II). *Klebsiella pneumonia* NY1 for municipal wastewater 72% suspended solids. *Paenibacillus elgii* B69 for municipal wastewater: 83% turbidity. *Paenibacillus mucilaginosus* for papermill wastewater: 81.5–88% suspended solids.	[55,57,107,108,109,110,111]
Novel sodium alginate (SA) supported tetrasodium thiacalix [4] arene tetrasulfonate (TSTC[4]AS-s-SA) nanogel (50 nm) and superparamagnetic nanocomposite of SA(Fe_3_O_4_@TSTC[4]AS-s-SA) was fabricated from coprecipitation of SA-supported tetrasodiumthiacalix[4]arene tetrasulfonate and in situ generated Fe_3_O_4_ nanoparticles.	Pb(II) > Cd(II) > Cu(II) > Cr(III) > Co(II) > Ni(II) at pH = 7. Adsorption capacity mg/g and % removal with TSTC[4]AS-s-SA: mg/g (%) Co 64.5 (12.9), Cd 89.14 (17.82), Pb 84.5 (16.9), Cu 87.82 (17.56), Ni 62.9 (12.58), Cr^3+^ 77.3 (15.46). Fe3O4@TSTC[4]AS-s-SA mg/g (%): 74.9 (14.98), 94.5 (18.9), 99.8 (19.96), 90.56 (18.11), 67.4 (13.48), 79.2 (15.48)	[112]
Novel adsorbent poly (methyl methacrylate)-grafted alginate/Fe_3_O_4_ nanocomposite by oxidative-free radical-graft copolymerization reaction.	62.5 mg/g Pb(II) and 35.71 mg/g Cu(II) at pH 5. Freundlich model at 50 °C.	[113]
Novel magnetic nanocomposite alginate beads, a3:4:1 aspect ratio (alginate: nanocomposite: xanthan gum) is used for fabrication of the beads.	The beads show removal percentage for phosphate at 97.9%, copper at 81.8%, and toluene at 43.4% and adsorption capacities of 60.24 mg/g, 120.77 mg/g and 25.52 mg/g, respectively. Isothermal studies show that the Langmuir isotherm model is the best governing equation for sorption. A pseudo-second-order model is the governing equation for the kinetics of sorption. The sorption process is also spontaneous and exothermic. The beads showed greater affinity in the order—${\mathrm{PO}}_{4}^{3-}$ > Cu^2+^ > toluene	[114]
PVA/SA beads via blending PVA with SA and the glutaraldehyde as cross-linking agent. The zeolite nanoparticles (Zeo NPs) were incorporated in the PVA/SA resulting in Zeo/PVA	99.5% Pb (II), 99.2% Cd(II), 98.8% Sr(II), 97.2% Cu(II), 95.6% Zn(II), 93.1% Ni(II), 92.4% Mn(II), 74.5% Li(II) for pH 6.0. 96.5% Fe(III), 94.9% Al(III) at pH 5 Natural wastewater samples: 60–99.8% of Al(III), Fe(III), Cr(III), Co(II), Cd(II), Zn(II), Mn(II), Ni(II), Cu(II), Li(II), Sr(II), Si(II), V(II), Pb(II).	[115]
Cobalt ferrite—alginate nanocomposite synthesized, ex situ polymerization	6.75 mg/g Reactive Red 195 and 6.06 mg/g Reactive Yellow 145 from a textile dye effluent in a binary component system	[116]
PVA/graphene oxide (GO)-SA nanocomposite hydrogel beads, in situ cross-linking, 0.15–0.2 μm.	279.43 mg/g Pb(II). Second-order kinetic model and Langmuir adsorption isotherm.	[117]
Alginate beads	107.53 mg/g Cu(II), 5 cycles of adsorption and desorption: 92% Cu(II).	[118]
Alginate/montmorillonite beads	Removal of Pb, with maximum of 244.6 mg/g at pH 6 and minimum of 76.6 mg/g at pH 1.	[119]
Alginate/Ag hydrogel, with Ag nanoparticles of 19 nm size	213.7 mg/g MB, Langmuir adsorption.	[120]
Cellulose/CuO nanoparticles	Microbial disinfection of waters: antibacterial activity against Gram-positive and Gram-negative bacteria.	[121]
CA/Fe nanoparticle membrane	99% CA—0.5% Fe nanoparticle blend ultrafiltration membrane applied for sulphates and organics removal, as biological oxygen demand (BOD) and chemical oxygen demand (COD) for textile industry effluent.	[92,122]
NH_2_-functionalized CA/silica composite nanofibrous membranes by sol-gel combined with electrospinning technology	19.46 mg/g as maximum adsorption capacity for Cr(VI)	[123]
TiO_2_/cellulose composite films by sol-gel method	Catalyst for phenol degradation	[92,124]
CA/Zinc oxide–Zeolite nanocomposite	Removal of Benzophenone-3, efficiency 98%	[125]
Lignocellulose/montmorillonite nanocomposite	Removal of Ni, 94.86 mg/g	[126]
Starch/Fe_3_O_4_	Removal of Pb^2+^, Cu^2+^, and Ni^2+^	[127]
Starch/polyaniline nanocomposite	Removal of Reactive Black 5, efficiency 99%, 811.3 mg/g	[92,128]
Chitosan/activated carbon/PVA (CS-AC-PVA) hybrid composite beads	Capacity of Pb^2+^ adsorbed was 0.2808 mg/g. The characteristics for Pb^2+^ adsorption process from aqueous environment were: - pH of 5 at temperature of 25 °C; - kinetics model followed pseudo-second-order kinetics, evidenced that Pb^2+^ ion was mainly adsorbed on the adsorbent surface via chemical interactions; - chemical adsorption was exothermic in nature.	[129]
Zr/Fe/Al-modified chitosan beads	Adsorption capacity of fluoride was 37.49 mg/g	[130]

**Table 3 nanomaterials-12-01707-t003:** Natural polymers and their nanocomposites used for soil remediation.

Type of Polymer or/and Nanocomposite	Soil Pollutants and Performances	References
Chitosan and PVA were added to alginate (10 wt.%) and cross-linked with epichlorohydrin (ECH)	70% adsorption efficiency, after 6 cycles of adsorption/desorption.	[151]
Nano-chitosan–urea composite encapsulation of urea with the chitosan polymer, 33.39 ± 11.84 nm, and 113.55 ± 19.02 nm chitosan	25% N as fertilizer required level as 75 kg N/ha recommended dose. Reducing with 3.36% ammonia-oxidizing bacteria (AOB) and 2.02% nitrate-reducing bacteria (NRB)	[152]
Chitosan–urea encapsulated persulfate for low-release synthesized by an emulsion cross-linking method	80% removal rate for pyrene in weakly acidic or neutral soil environments	[153]
Novel chitosan/clay/biochar nanobiocomposite. Biochar mesopores (pores 2–50 nm) and mean pore diameter: 1.9842 nm.	121.5 mg/g Cu, 336 mg/g Pb, and 134.6 mg/g Zn. Synthetic precipitation leaching procedure: 10 g soil with 10% nanobiocomposite in synthetic rain water (20 g/L), 24 h. Freundlich model for Cu(II) and Zn(II) and Temkin model for Pb(II). Immobilization: 100% (Cu), 100% (Zn), and 52.29% (Pb).	[154]
Carboxymethyl cellulose (CMC) support for montmorillonite-stabilized iron sulfide composite	90.7% Cr(VI) after 30 days, with 5% (composite–soil mass proportion), measured using the toxicity characteristic leaching procedure.	[155]
CMC—nanozerovalent iron (CMC-nZVI), with 80–120 nm nZVI	Leachability: 100% Cr(VI) and 95.8% Cr total, with 2.5 g/Kg CMC-nZVI. Immobilization: 45.4% Crtotal and 17.9%Cr(VI) with 1 g/kg; 72.8% Crtotal and 58.6% Cr(VI) with 2.5 g/kg; 95.8% Crtotal and 100% Cr(VI) with 5 g/kg.	[156]
CMC bone-char/CMC stabilized FeS composite = 1:1:1	452.99 mg/g, pH: 2.0–6.0, 65.47%.	[156]
Alginate for composite powder: Fe-AC-alg	Over 96% efficiency with 1g composite for polycyclic aromatic hydrocarbons (PAHs) (anthracene (Ant), phenanthlene (Phe), fluoranthene (Flu), pyrene (Py), benz[a]anthrathene (BaA), chrysene (Chr), benzo[b]fluroranthene (BbF), benzo[k]fluroranthene (BkF), benzo[a]pyrene (BaP), dibenz[a,h]anthracene (DahA), benzo[ghi]perylene (BghiP), and indeno(1,2,3-cd) pyrene (IP)). Low recovery (%) for Py (24), BaA (43), Chr (5.4), and BbF (3.9%). Other PAHs were not recovered.	[157]
Alginate spheres with magnetic hollow carbon composite	44.02% Cd removal with 1.5 g composite/60 g soil. Composite recyclability: 88.87% in flooding soil and 94.45% in non-flooding soil.	[158]
SA gel beads incorporated silicon sulfuretted nanoscale zero valent iron (FeSSi) with specific surface 101.61 m^2^/g	Removal efficiency: 80.10% (Cd), 99.96% (Pb), 66.80% (Ni), and 80.46% (Cr) with pseudo-second-order model. Leaching tests for recovery rate (Rr) of heavy metals from solution (Rr/w) and soil (Rr/s): 59.79–98.70% and 25.94–62.67% with 0.3 g SA-FeSSi.	[159]
Lignin, CMC, and SA amendments	Leaching concentrations: 5.46–71.1% and 4.26–49.6%, 1.0 g of soil, pH 2.88, 18 h, 30 rot/min.	[160]

**Table 4 nanomaterials-12-01707-t004:** Main performances of natural polymers as filtration substrates for air decontamination.

Pollutant	Type of Polymer	Performances/Mechanism	References
PM 2.5 and 10 μm	Uniform silk protein nanofibers by electrospinning process	Air filtration efficiencies: 90% and 97%, exceeding the performances of commercial semi high-efficiency particulate air (semi-HEPA) filters. Nanofibers are naturally degraded.	[217]
PM 10 μm, including aerosol particles: DEHS (diisooctyl sebacate particles as organic particle matter) and NaCl (sodium chloride particles as inorganic matter)	Chitosan/PVA nanofibers with SiO_2_/Ag nanoparticles as air filtration nanofibrous membrane	Filtration efficiency: 96% for particles between 300 nm–1 μm and 100% for micron level particles. Composite membrane weights: between 1.48 and 6.2 g/m^2^ for filtration efficiency: NaCl particles from 42.97% (pressure drop is 33.67) to 96.60% (pressure drop is 305.67); DEHS particles from 51.01% (pressure drop: 33.67) to 99.12% (pressure drop: 296.17).	[178]
PM 2.5 and 10 μm	Biodegradable electrospun PLLA polymer nanofibers for air filter applications. polymer nanofibers are ≈500 nm	Efficiency: 99.3%. Even after 6 h of filtration time, the PLLA filtration membrane still exhibits a 15% improvement in quality factor for PM 2.5 particles compared to the 3M respirator. Similarly for PM 10 particles, these quality factors of the (poly(D-lactic acid)) PDLA and poly(L-lactic acid)PLLA membranes exhibited 3% and 4.6% improvements compared to the 3M respirator after 6 h filtration time. Furthermore, the PLLA filter membrane also exhibited a high porosity of 91.9%, a specific surface area of 4.5 m^2^/g, and a dust-holding capacity of 7.36 g/m^2^.	[219]
PM 2.5	The average diameter of the electrospun nanofibers used was 239 nm, ranging from 113 to 398 nm.	Aerosol particles (diameters from 7 to 300 nm). Experimental results indicated that the nanofibers showed good permeability (10−11 m^2^) and high-efficiency filtration for aerosol nanoparticles (about 100%), which can include BC and the new coronavirus. The pressure drop was 1.8 kPa at 1.6 cm/s, which is similar to that reported for some high-efficiency nanofiber filters. In addition, it also retains BC particles present in air, which was about 90% for 375 nm and about 60% for the 880 nm wavelength. Additionally, nanofiber retention efficiencies for atmospheric PM 2.5 and BC were analyzed.	[43]
PM 0.3, PM10	ZnO@PVA/konjac glucomannan (KGM) membranes gelatin nanofiber, areal density of 3.43 g/m^2^	ZnO@PVA/KGM filtration efficiency: 99.99% for ultrafine particles with the size of 300 nm. Gelatin nanofiber filtration efficiency: 99.3% (PM 0.3) and 100% (PM 2.5)	[189,219]
PM 2.5, *Escherichia coli*	Soy protein isolate (SPI)/PVA electrospinning membrane	Filtration efficiency: 99.99% for PM < 2.5 µm and inhibiting effect on *Escherichia coli*	[220]

## Data Availability

No new data were created or analyzed in this study. Data sharing is not applicable to this article.

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
