# Peer review of "Natural Polymers and Their Nanocomposites Used for Environmental Applications"

_nanomaterials, 2022, doi:10.3390/nano12101707_

Round 1

Reviewer 1 Report

The manuscript focuses on natural polymers and their nanocomposites as remediation materials application in water, soil and air pollution. It describes very well the topic but lacks of much more opinions from the authors that should be added.

Author Response

Thank you for your appreciation!

 Some opinions of authors are provided along of manuscript, for example in Conclusion section:

“Based on our investigations, we observed that the combination of a natural component with a nanosized particle lead to development of innovative materials with a real capture potential of the target pollutant from combined water soil systems.”

“Our study demonstrates the advantages of using nanocomposites through the dual functionality of the two components (nanoparticles and polymer) which offer advanced properties such as: specific surface area, reactivity, stability.”

Reviewer 2 Report

The manuscript "Natural Polymers and their Nanocomposites used for Environmental Applications" is a review on the application of biopolymers and composite biopolymers for water, soil, and air decontamination. This is a very hot topic and this review can be appealing for the scientific community. Moreover, the work is well organized and written. Therefore, the publication is recommended; but after some revisions.

Detailed comments:

  • Abstract. Add the main properties of the biopolymers to be suitable for this application and the main conclusions of the work;
  • Add an abbreviation list to help the reader;
  • Introduction. Describe the environmental problem related to organic pollutans of water, soil and air. For this purpose, see these recent works: Somma et al., ChemEngineering2021, 5(3), 47; Gul Zaman et al., Materials, 2021, 14(24),7607; Ojstrsek et al., Materials, 2022, 15(4), 1488; etc. Extent the aim of the review, highlighting the novelty of the work and the features of natural polymers with respect to the synthetic ones; Specify the reasons for the selection of those biopolymers (chitosan, cellulose, PLA, etc..);
  • Underline the morphological and mechanical properties required for the biopolymers to be used for this specific application;
  • See journal template for text and tables organization;
  • Conclusions. Include the optimal systems (kind of polymer, morphology, etc..) identified for water, soil and air remediation.
  • Correct typos.

Author Response

Thank you for your appreciation!

We modified the abstract as follows:

“The aim of this review is to bring together the main natural polymer applications for environmental remediation, as a class of nexus materials with the advanced properties that offer the opportunity of integration in single or of simultaneous decontamination processes. By identifying the main natural polymers derived from agro-industrial sources or monomers converted by biotechnology into sustainable polymers, the paper offers the main performances identified in the literature for: (i) treatment of contaminated waters with heavy metals and emerging pollutants such as dyes and organics, (ii) decontamination and remediation of soils and (iii) reduction of the number of suspended solids of particulate matter (PM) type in the atmosphere. Because nanotechnology offers new horizons in materials science, nanocomposite tunable polymers are also studied and presented as promising materials in the context of developing sustainable and integrated products in society to ensure quality of life. As a class of future smart materials, the natural polymers and their nanocomposites are obtained from renewable resources being as inexpensive materials with high surface area, porosity and high adsorption properties due to their various functional groups. The information gathered in this review paper are based on the publications in the field from the last two decades. Future perspectives of these fascinating materials should take into account the scale-up, toxicity of nanoparticles, competition with food production, as well as environmental regulations.”

A list of abbreviations was provided inside of manuscript.

We added in the introduction new information from the literature research as follows:

“Based on our study, the published researches from the last 20 years have focused on the retention of heavy metals as target pollutants in water and their immobilization in soil.   With regard to air as environmental factor, the literature offers solutions especially for the retention of particulate matter (PM). To a lesser extent, the studies have also focused on organics removal. All this information presented in this review paper has been classified according to the importance derived from the data analyzed.”

Regarding organics environmental problem, we studied the suggested references and included them in chapter 4. We considered that this information is more suitable in this chapter according to the other reviewers’ suggestions:

“Besides heavy metals, waters could contain a considerable organic pollutant quantities with high impact on life quality. Some of these emerging pollutants rise serious threatens due to their toxicity and lack of regulations. The organic pollutants with the greatest impact on the environmental factors are: dyes, pharmaceutically active compounds, endocrine-disrupting chemicals, personal care products, and flame retardants [87].

In accordance with the required legislation limits, the water management focuses on improvement of the aquatic ecosystems quality and environmental protection even if official limits for these type of pollutants are not still available [88].

Together with these, polycyclic aromatic hydrocarbons (PAHs), persistent organic pollutants (POPs), especially dioxins, furans, chlorinated pesticides (OCP) and polychlorinated biphenyls (PCBs) are also monitored both in water and soil being present on a large scale in ecosystems due to the human activities [89].

There are various treatment methods for the organics removal, among them the most used being adsorption by help of nanomaterials, nanocomposites, nanoparticles, clays, biopolymers, metal–organic frameworks (MOFs), and zeolites [90].”

“The novelty of this paper is to emphasize the main performances obtained when natural polymers are used as matrix for developing of advanced nanocomposites for environment decontamination”.

“According to the literature researches, the most promising and commonly reported natural polymers, as single or in combination with other advanced materials are chitosan, alginate, cellulose, lignocellulose, starch, and PVA.”

“In addition to medical applications, environmental factors could also be remediated by use of these natural materials. It is well-known that sustainable materials are usually derived from synthetics for their durability but this causes a great threaten to the environment due to their harmful chemicals. In the recent last years, it has been shown that the replacing of at least one synthetic component with natural one could enhance environmental protection with-out compromising the original     maintaining the initial properties of these materials. In this regard, our new approach sustains the use of these natural polymers as viable alternative for a clean and safe environment.

Based on our study, the published researches from the last 20 years have focused on the retention of heavy metals as target pollutants in water and their immobilization in soil.  With regard to air as environmental factor, the literature offers solutions especially for the retention of particulate matter (PM). To a lesser extent, the studies have also focused on organics removal. All this information presented in this review paper has been classified according to the importance derived from the data analyzed. Even if this review is restricted to articles published in the last two decades, there are also presented older but relevant data related to the natural polymers.”

Generally, the mechanical properties of biopolymers are weaker and their improvement is obtained by combining with other NPs. The Introduction was completed with the following text:

“The main properties of biopolymers to be used as filter membranes or adsorbents for environmental applications are related to the high permeability to capture airborne or water contaminants [9], outstanding  mechanical and thermal properties [9], better resistance to diverse pH conditions [10], biodegradability, swelling capacity, slight modification to increase the adsorption, and last but not last, to have a low cost [11]. For processing of fibers by electrospinning tehnnology, the specific features of biopolymers are needed as an adequate solubility and compatibility in aqueous solvents, a certain molecular weight, a desired surface topography to attract positively or negatively ions. All biopolymer’s requirements will permit to obtain nanometer-sized fibers with controlled fiber diameter, without bead defect and specific surface area [12].”

The manuscript was checked again and the text and tables organized according to the journal template.

"In conclusions, we highlighted the recent performances regarding the use of natural polymers and polymeric nanocomposites especially in eliminating and/or immobilization of heavy metals, and subsidiary organics from soil and water.

Enormous environmental threatens, such as climate changes due to the carbon release, waste disposals, water and air quality force society to find sustainable solutions for life quality. Besides, the well-known concepts such as sustainable development combine with circular bio-economy has to be implemented such that biodiversity not to be affected and future generations to have a stable and clean environment.

Available natural resources could be used as the next-generation of the advanced materials with targeted applications. Also, agro-industrial biomass based on natural substances such as polymers or mixtures of them or biotechnology applied for monomer production offer interesting natural structures that could be tunable for enhanced properties. It is well-known the application of natural polymers for biomedical, pharmaceutical and food industry. Last years, researches in environmental remediation, especially for soil, indicate promising results with natural polymers as single or nanocomposites, especially with chitosan. This paper integrates the most relevant results for water, soil and air systems when natural polymers and their nanocomposites are applied as remediation materials. Based on our investigations, we observed that the combination of a natural component with a nanosized particle lead to development of innovative materials with a real capture potential of the target pollutant from combined water soil systems. The actual result proves the efficiency for heavy metals removal and opens new perspectives for the removal of organics based on the preliminary results. Our study demonstrates the advantages of using nanocomposites through the dual functionality of the two components (nanoparticles and polymer) which offer advanced properties such as: specific surface area, reactivity, stability. In addition, the natural polymer as green compound has the advantage of availability and low cost. A more comprehensive vision for the future should be centered on scale-up commercial and industrial applications. This results from the proven laboratory efficiencies combined with more and more environmental regulations."

The manuscript was revised and corrected for typos.

Reviewer 3 Report

The authors have submitted a review " Natural Polymers and their Nanocomposites used for Environmental Applications, " highlighting publications related to applying biopolymer-based nanosystems for environmental purposes. The manuscript is well structured and reads well overall, although it will need a spelling check. However, some important parts should be revised, added, and discussed in the manuscript. Finally, I suggest this article be published after a major revision.

 The abstract is clear and concise.

The introduction is clear and concise.

The various sections in the body text of the review are clear and concise.

Comments:

1- First of all, I would like to recommend authors to design a “Graphical Abstract” for this study to better show the whole story in a simple and informative manner.

2- There are several misspellings and errors in the style which should be revised. The manuscript should be double-checked in terms of English and style.

Please keep consistency in your writing style in the whole manuscript (including figures and tables). For example, in the manuscript, we can see both nano-composite and nanocomposite.

3- Figure 2 is messed up. The Figure is not clear and reactions are mixed with the SEM image and there are not readable. Please resketch it. Same comment for the next figure. The texts are not legible.

4- The alteration of nanoparticles’ characterization in terms of size, morphology, and topology after and composition and the final synergistic impacts of the composition of nanoparticles and biopolymers can better be explained with the addition of one or two figures to the manuscript from published papers in the field. In this regard, you can explain the benefits of this type of composition from both material and environmental perspectives to the readers.

5- The most interesting and important types of nanoparticles that have been dramatically exploited for environmental purposes (air and water purification) are Metal-organic frameworks (MOFs) and their subgroups (ZIFs, MILs, etc.). There are many articles out there related to the composition of MOFs and natural polymers that I believe a special section should be considered for this part. Naturally found polymers such as eggshell membrane and pullulan also should be added since there are a plethora of articles based on this biomaterial. Please explain about MOF/Natural Polymer nanocomposites. Similarly for zeolite-based composites. Please check out the following articles in the field :

Applied Surface Science, 480, pp.288-299, Carbohydrate polymers 227 (2020): 115364, Environmental Technology & Innovation, 23, p.101747, Carbohydrate Research, 489, p.107930

6- Natural polymers in terms of membrane and more specifically hydrogel membrane environmental application also should be added to the manuscript. You find interesting information in the following reference: Materials Science and Engineering: C 114 (2020): 111023

7- Some of the references in this paper are so old (1986, 2005, etc.) and only one cited paper is for 2022. Please update citations and try to use new papers in your review.

8- Conclusion is too short. Future trends of these nanocomposites are also expected to be mentioned.

Author Response

Thank you for your appreciation!

A spelling check was carried out along all manuscript.

  1. The Graphical Abstract was elaborated and uploaded.
  2. In this form, the manuscript was corrected from the misspellings and errors in the style and also from English language. 
  3. Figure 2 (new Fig. 4) and Figure 3 (new Fig. 5) were changed with other ones, we think with a better quality.
  4. Section 3 was completed with:

    “It is well recognized that the toxicity of nanoparticles as well as the methods used for their synthesis and modification with natural polymers can have a negative impact of the environment.

    The impact of the nanoparticles is larger than that of the polymer matrices. Adding of NPs to the natural polymer matrix permits to obtain nanocomposite polymeric materials having beneficial for environment related to the replace of the petroleum resources that are limited, valorization of sustainable resources for obtaining of natural plastics and also, the creation of the performing properties for different environmental applications [65]. For example, the chitosan/silver nanocomposite performed enhanced antimicrobial activity than chitosan itself in waste water and decolourisation of methyl orange (MO) – Fig. 2.”

    In Section 3 Figure 2. Eco-friendly approach of chitosan/silver nanocomposite used for dye removal for potable water [66] (with permission of Elsevier) and Figure 3. The performance of MOF for air filtering media [9] were added.

    “Synergetic impact between the chitosan with zeolite nanoparticles led to a biocompatible mesoporous networks material with low toxicity, improved mechanical properties, narrow pore size distributions and high surface areas features as adsorbent of water pollutants.”
  5. Thank you for your suggestions and indicated article papers!

    We completed the Section 3 with newly Fig. 2 and Fig. 3 and the following text:

    “Chitosan-modified zeolite composites are a promising platform for  environmental engineering applications [69]. Synergetic impact between the chitosan with zeolite nanoparticles led to a biocompatible mesoporous networks material with low toxicity, improved mechanical properties, narrow pore size distributions and high surface areas features as adsorbent of water pollutants.

    Inspired by natural zeolites, the most interesting and important types of nanoparticles that have been dramatically exploited for environmental purposes (air and water purification) are metal-organic frameworks (MOFs) and their popular subclasses (zeolitic imidazolate frameworks (ZIFs), Materials of Institute Lavoisier frameworks (MILs), etc.). MOFs are wonderful materials widely studied for removing contaminants from the effluents [70]. They are characterized by nontoxicity, high chemical stability, high adsorption capacity, large porosity, high inner surface area, ultrahigh thermal and chemical stabilities [69]. Nanocellulose-based filtering materials were developed by incorporation of ZIF nanocrystals in the cellulose microfibers network, as a subclass of MOF HKUST-1 [9] – Fig. 3.

    “The addition of ZIF-8 nanocrystals to cellulose microfibers led to the improved surface area proved by an excellence enhance in the BET surface area from 6.66 up to 620.80 m2/g compared with unmodified surface, which led to the increase in the filtration efficiency from 99.5 to 99.9% against PM 0.3 particles [9]. ZIF-8 crystal was found to coated the  surface of chitosan/polyvinyl alcohol electrospun nanofiber (CS/PVA-ENF) for dye removal from wastewater treatments [71]. In this case, the high adsorption capacity of 1000 mg/g was recorded during the second cycle.

    In another paper, zeolitic imidazolate frameworks-67 (ZIF-67) crystals were incorporated on the surface of magnetic eggshell membrane (Fe3O4@ESM) resulting the ZIF-67@Fe3O4@ESM composite as a novel adsorbent with the high surface area of 1263.9 m2/g [70]. Maximum adsorption capacity of 344.82 and 250.81 mg/g for Cu(II) and Basic Red 18 (BR18) dye, respectively were reported [70]. The advantage of use of this adsorbent is that the magnetite favors the facile separation from aqueous media. Recently, MILs have attracted a much research attention as promising adsorption for heavy metal removed from wastewaters [70]. The advantage of use of this adsorbent is that the magnetite favors the facile separation from aqueous media.

    Recently, Materials of Institute Lavoisier frameworks (MILs) have attracted a much research attention as promising adsorption for heavy metal removed from wastewaters [72].”
  6. Thank you for your suggestion!

    We completed the Section 3 with following:

    “Natural polymers and their combinations are successfully used for creation of hydrogel membranes (HMs), a crosslinked-three dimensional (3D) porous structure that can act as adsorbents for heavy metals or organic contaminants from treatment wastewater. For example, HMs based on PVA for adsorption of strontium ions from wastewater treatment, calcium alginate (CaAlg) coated with iron nanoparticles (Fe NPs, ~5 nm) having a 99.5% efficacy to remove Cr(VI) from contaminated water [73], magnetic chitosan cross-linked with glyoxal (Fe(3)O(4)NPs/CS/glyoxal) for removal 80-90% of Cr(VI) from water [74], carboxymethyl cellulose (CMC) g-poly(2-(dimethylamino) ethyl methacrylate) (CMC-g-PDMAEMA) to remove MO from aqueous water with adsorption capacity of 1825 mg/g [75] are reported”.
  7. We updated the references with article from the last two decades, but some data related to natural polymers selected for review are limited, important and relevant to the topic of the article even if they are from older articles as it is observed in the Introduction:

    “Based on our study, the published researches from the last 20 years have focused on the retention of heavy metals as target pollutants in water and their immobilization in soil.   With regard to air as environmental factor, the literature offers solutions especially for the retention of particulate matter (PM). To a lesser extent, the studies have also focused on organics removal. All this information presented in this review paper has been classified according to the importance derived from the data analyzed. Even if this review is restricted to articles published in the last two decades, there are also presented older but relevant data related to the natural polymers.”
  8. Conclusions were modified as follows:

     “In conclusions, we highlighted the recent performances regarding the use of natural polymers and polymeric nanocomposites especially in eliminating and/or immobilization of HMs, and subsidiary organics from soil and water.

    Enormous environmental threatens, such as climate changes due to the carbon release, waste disposals, water and air quality force society to find sustainable solutions for life quality. Besides, the well-known concepts such as sustainable development combine with circular bio-economy has to be implemented such that biodiversity not to be affected and future generations to have a stable and clean environment.

    Available natural resources could be used as the next-generation of the advanced materials with targeted applications. Also, agro-industrial biomass based on natural substances such as polymers or mixtures of them or biotechnology applied for monomer production offer interesting natural structures that could be tunable for enhanced properties. It is well-known the application of natural polymers for biomedical, pharmaceutical and food industry. Last years, researches in environmental remediation, especially for soil, indicate promising results with natural polymers as single or nanocomposites, especially with chitosan. This paper integrates the most relevant results for water, soil and air systems when natural polymers and their nanocomposites are applied as remediation materials. Based on our investigations, we observed that the combination of a natural component with a nanosized lead to development of innovative materials with a real capture potential of the target pollutant from combined water soil systems. The actual result proves the efficiency for HM removal and opens new perspectives for the removal of organics based on the preliminary results. Our study demonstrates the advantages of using nanocomposites through the dual functionality of the two components (nanoparticles and polymer) which offer advanced properties such as: specific surface area, reactivity, stability. In addition, the natural polymer as green compound has the advantage of availability and low cost. A more comprehensive vision for the future should be centered on scale-up commercial and industrial applications. This results from the proven laboratory efficiencies combined with more and more environmental regulations.”

Reviewer 4 Report

Matei et al. have reviewed on natural polymers and their nanocomposites used for environmental applications. It is a timely review that covers important natural polymers and their nanocomposites. The findings and results from various key articles are discussed and tabulated. The language and presentation were good. However, the following points need to be addressed before this paper could be accepted for publication.

  1. Abstract – the abstract is very simple and needs to be rewritten based on the discussed trends, correlations, associations and attributions highlighting the key striking reports. Also, a sentence summarizing the future perspective should be added at the end.
  2. The authors should mention if they have restricted their review with articles published within certain time limits.
  3. Keywords -
  4. Figure 1 – avoid using abbreviations alone, for instance PLA, PM, AOB, NTB. Instead provide the full form.
  5. Table 4 – “air depollution” should be replaced with “air decontamination” to be consistent with usage in other places.
  6. Conclusion – the conclusion is more general and simple. It should summarize the important trends and key striking reports discussed in this study. The future perspective is missing and it should be added in the conclusion. Change the title as “Conclusion and future perspective/outlook.
  7. The values and the units, unit formatting, superscript and subscript formatting, abbreviation for seconds, minutes, hour, day, month as s, min, h, d, mo, all of these should be double-checked for correctness throughout the manuscript.
  8. Care should be taken ensure if all the abbreviations are explained in full form at the first instance and abbreviated thereafter in both text, tables and figures.
  9. References – some important closely-related papers related to this review topic should be cited such as   https://doi.org/10.3390/membranes11020139
    https://doi.org/10.1016/j.ijbiomac.2020.07.244
    https://doi.org/2147/IJN.S34396

Author Response

Thank you for your appreciation!

1. We updated the abstract as follows:

“The aim of this review is to bring together the main natural polymer applications for environ-mental remediation, as a class of nexus materials with the advanced properties that offer the opportunity of integration in single or of simultaneous decontamination processes. By identifying the main natural polymers derived from agro-industrial sources or from monomers converted by biotechnology into sustainable polymers, the paper offers the main performances identified in the literature for: (i) treatment of contaminated waters with heavy metals and emerging pollutants such as dyes and organics, (ii) decontamination and remediation of soils and (iii) reduction of the number of suspended solids of particulate matter (PM) type in the atmosphere. Because nanotechnology offers new horizons in materials science, nanocomposite tunable polymers are also studied and presented as promising materials in the context of developing sustainable and integrated products in society to ensure quality of life. As a class of future smart mate-rials, the natural polymers and their nanocomposites are obtained from renewable resources being as inexpensive materials with high surface area, porosity and high adsorption properties due to their various functional groups.  The information gathered in this review paper are based on the publications in the field from the last two decades. Future perspectives of these fascinating materials should take into account the scale-up, toxicity of nanoparticles, competition with food production, as well as environmental regulations.”

2. We added in Introduction section the suggested information:

“Even if this review is restricted to articles published in the last two decades, there are also presented older but relevant data related to the natural polymers.”

3. The keywords were already introduced:

Keywords: polymers; nanocomposites; heavy metals; PM; soil remediation.

4. Figure 1 was changed according to the reviewers’ suggestion.

5. We made the change in Table 4 according to your suggestion.

6. "In conclusions, we highlighted the recent performances regarding the use of natural polymers and polymeric nanocomposites especially in eliminating and/or immobilization of heavy metals, and subsidiary organics from soil and water.

Enormous environmental threatens, such as climate changes due to the carbon re-lease, waste disposals, water and air quality force society to find sustainable solutions for life quality. Besides, the well-known concepts such as sustainable development combine with circular bio-economy has to be implemented such that biodiversity not to be affected and future generations to have a stable and clean environment.

Available natural resources could be used as the next-generation of the advanced materials with targeted applications. Also, agro-industrial biomass based on natural sub-stances such as polymers or mixtures of them or biotechnology applied for monomer production offer interesting natural structures that could be tunable for enhanced properties. It is well-known the application of natural polymers for biomedical, pharmaceutical and food industry. Last years, researches in environmental remediation, especially for soil, indicate promising results with natural polymers as single or nanocomposites, especially with chitosan. This paper integrates the most relevant results for water, soil and air systems when natural polymers and their nanocomposites are applied as remediation materials. Based on our investigations, we observed that the combination of a natural component with a nanosized lead to development of innovative materials with a real capture potential of the target pollutant from combined water soil systems. The actual result proves the efficiency for heavy metals removal and opens new perspectives for the removal of organics based on the preliminary results. Our study demonstrates the advantages of using nanocomposites through the dual functionality of the two components (nanoparticles and polymer) which offer advanced properties such as: specific surface area, reactivity, stability. In addition, the natural polymer as green compound has the advantage of availability and low cost.

A more comprehensive vision for the future should be centered on scale-up commercial and industrial applications. This results from the proven laboratory efficiencies combined with more and more environmental regulations."

7. The manuscript was double-checked for errors in unit formatting, superscript and subscript formatting, typos and abbreviations. The necessary modifications were made according to the reviewer’s suggestions.

8. The abbreviations were explained in full form at the first instance and just the abbreviations used thereafter. Also a list of abbreviations was provided.

9. We added the first two references [76] and [28] according to your suggestions. The last one was inaccessible.

Round 2

Reviewer 2 Report

The manuscript has been improved, according to the Reviewer's suggestions.

Reviewer 3 Report

The manuscript is well revised and is ready to be published. I have no further comments.